# A Multimodal Intervention for Prevention of Overweight and Obesity in Schoolchildren. A Protocol Study “PREVIENE-CÁDIZ”

**DOI:** 10.3390/ijerph18041622

**Published:** 2021-02-08

**Authors:** Rubén Aragón-Martín, María del Mar Gómez-Sánchez, David Jiménez-Pavón, José Manuel Martínez-Nieto, Mónica Schwarz-Rodríguez, Carmen Segundo-Iglesias, José Pedro Novalbos-Ruiz, María José Santi-Cano, José Castro-Piñero, Carmen Lineros-González, Mariano Hernán-García, Amelia Rodríguez-Martín

**Affiliations:** 1Department of Biomedicine, Biotechnology and Public Health, Faculty of Nursing and Physiotherapy, University of Cádiz, 11009 Cádiz, Spain; josepedro.novalbos@uca.es (J.P.N.-R.); amelia.rodriguez@uca.es (A.R.-M.); 2European ITI Project PI-0007-2017, Andalusian Operational Program FEDER (European Regional Development Fund) 2014–2020, 11009 Cádiz, Spain; david.jimenez@uca.es (D.J.-P.); josemanuel.martinez@uca.es (J.M.M.-N.); monica.schwarz@uca.es (M.S.-R.); csegundoiglesias@gmail.com (C.S.-I.); mariajose.santi@uca.es (M.J.S.-C.); jose.castro@uca.es (J.C.-P.); carmen.lineros.easp@juntadeandalucia.es (C.L.-G.); easp15@gmail.com (M.H.-G.); 3Biomedical Research and Innovation Institute of Cádiz (INiBICA) Research Unit, Puerta del Mar University Hospital, University of Cádiz, 11009 Cádiz, Spain; 4MOVE-IT Research Group, Department of Physical Education, Faculty of Education Sciences, University of Cádiz, 11003 Cádiz, Spain; 5Department of Nursing and Physiotherapy, Faculty of Nursing and Physiotherapy, University of Cádiz, 11009 Cádiz, Spain; 6Salus Infirmorum Nursing School, University of Cádiz, 11001 Cádiz, Spain; 7GALENO Research Group, Department of Physical Education, Faculty of Education Sciences, University of Cádiz, 11003 Cádiz, Spain; 8Andalusian School of Public Health, 18080 Granada, Spain; 9Andalusian Council of Childhood, 18001 Granada, Spain

**Keywords:** pediatric obesity, primary prevention, health behavior, eating behavior, physical activity, sedentary behavior, sleep, multimodal intervention

## Abstract

This paper describes the protocol for a study designed to address the high prevalence (40%) of childhood overweight and obesity in the province of Cádiz, Spain, as a reflection of what is happening worldwide. It is widely known that children who suffer from childhood obesity have a higher risk of developing chronic diseases in adulthood. This causes a decrease in the quality of life and an increase in health spending. In this context, it is necessary to intervene promoting healthy lifestyle habits from an early stage. The objective of this project will be to evaluate the effectiveness of a multimodal intervention (individual, school and family) called “PREVIENE-CÁDIZ” [CADIZ-PREVENT]. The intervention will be focused mainly on diet, physical activity, sedentary lifestyle and sleep, to prevent overweight and obesity in schoolchildren from 8 to 9 years old in the province of Cádiz. It will consist of a 10-session education program carried out in the classroom by the teachers. In addition, children will be assigned two workbooks, one to work on in class and the other at home with parents. A workshop aimed at parents will be included to help teach them how to obtain healthier lifestyle habits. The proposed study will involve a quasi-experimental design with a control group.

## 1. Introduction

The World Health Organization (WHO) considers obesity to be a problem or main issue of concern for public health today in the developed world, due to the high prevalence of this disease in these countries and all the problems it causes. Together with the International Obesity Task Force (IOTF), obesity has been classified as the epidemic of the 21st century [1,2]. The prevalence of this epidemic has increased in Spain by three to four times in the last 20 years and has not only caused a strong impact on chronic diseases, but also on health spending and the quality of life of the population [3]. Moreover, estimates have been made [4] that by 2050, obesity rates in countries of the Organization for Economic Co-operation and Development (OECD) may affect 60% of men, 50% of women and 25% of children.

In a recent study that evaluated the levels of severe obesity in child populations (6–10 years) of 21 European countries, a conclusion was made that it presented a major public health problem for a large number of countries, especially relevant in Bulgaria, Greece, Italy and Spain. This higher prevalence of obesity in southern European countries has been demonstrated in previous research [4,5], estimated at more than 40%. When the prevalence of obesity in these countries is studied specifically (Italy [6], Greece [7] or Spain), the data are confirmed. Particularly in Spain, the Aladdin study [8], which was carried out between November 2015 and March 2016, found for this age group a percentage of 39.3% with overweight or obesity. Of this percentage, 23.2% suffered overweight and 16.1% obese. The obesity was more frequent in boys than girls.

Within Spain, the province of Cádiz (Southwest) has the highest prevalence of overweight and obesity (35% children aged 6–12 years) compared to the rest of Spain, which constitutes a serious health problem in the province that has been increasing in recent years [9]. Researchers from the Andalusian School of Public Health [10] reported a similar prevalence of overweight and obesity (37%) in children aged 8–9 years in Granada (Southeast Spain). This data confirms the need to develop and reinforce public health policies that reduce these numbers.

In the general population, obesity increases the risk of numerous diseases such as diabetes, high blood pressure, dyslipidemia, cardiovascular diseases, some types of cancer, musculoskeletal disease, obstructive sleep apnea, and fatty liver disease [11,12,13,14]. Life expectancy for 40 year old people who suffer obesity is seven years less than that of one without excess weight [15]. Similarly, obesity in children and adolescents is associated with a higher prevalence of various disease risk factors, such as pre-diabetes, increased blood pressure, accumulation of cholesterol in the arteries, increased oxidative vulnerability and sleep disorders [16,17,18,19,20], increased risk of excess body weight and related diseases in adulthood [21,22]. Excess body weight is also associated with lower self-esteem and higher levels of anxiety and depression, thus reducing well-being and quality of life [20]. Similarly, a study carried out in a sample of children aged 2–9 years in eight European countries found that children who were overweight had an increased risk of poor health-related quality of life.

Due to all the problems associated with obesity that have been described in the previous paragraph, it is urgent to detect and solve this disease in the early stages of life, both to protect the general health of the population as well as to promote lifelong engagement of healthy lifestyle behaviors [23]. Children mostly depend on their family, school, health services or social institutions to acquire relevant knowledge and skills to grow into healthy adults [24]. The important influence of the environment that surrounds the child (socioeconomic, educational level, etc. of the parents) on the healthy habits of the child has been shown in numerous studies [25,26,27]. The study conducted by Spinelli et al. [25], shows that children with parents with low and medium levels of education constitute a significant risk to suffer overweight or obesity. Children whose mothers had only primary or secondary levels of education, had a higher prevalence of severe obesity than children whose mothers had high levels of education. In the province of Cádiz, the sociocultural level is low for 25% of the adult population, as reflected in a study carried out by this research group, where the educational, employment and socioeconomic level of the adult population from the province of Cádiz was analyzed [28]. Due to this, an educational intervention based on health promotion is proposed, since this type of interventions allows individuals to overcome these inequalities, reaching children of all socioeconomic levels.

A Cochrane review [29] analyzed 37 studies focused on boys and girls between the ages of 6 and 12 years old. They evaluated interventions focused on behavior change in school settings. They concluded that the impact of the interventions translated into a reduction in BMI of 0.15 kg/m^2^ in the aforementioned age group. The authors recommended that the school curriculum was reinforced, including content on healthy eating, nutritional balance, a higher level of physical activity and personal self-image. In addition, they claimed greater support from educational personnel to achieve the goals.

It has been concluded in previous studies that the school environment is the most appropriate place for the development of strategies aimed at reducing childhood overweight/obesity, especially when the family is involved [30]. Interventions in school settings have been suggested to represent a marked improvement in terms of public health, and future interventions involving children and parents jointly are recommended [31].

Also, the duration and design of the study have also been considered in various studies, suggesting a minimum duration of six months to assess changes in lifestyle habits [31], thus our intervention will last six months. In addition, performing the intervention in a school year will allow the sample to be obtained and minimize the possibility of abandonment, as previously described [32,33]. Similarly, quasi-experimental designs or clinical trials have been suggested as the most appropriate approaches for this type of study [34].

Martínez-Vizcaíno et al. [35] have shown changes in the proportion of body fat in nine-year-old schoolchildren by means of the implementation of an extracurricular program of non-competitive physical activity. Ávila-García et al. [36] analyzed interventions for the promotion of eating habits and physical education in Spanish primary school students, identifying only seven publications that met the inclusion criteria, which shows the scarcity of research in this area. In this review, it was concluded that half of the studies showed positive changes in the improvement of body composition, and all had a positive effect regarding eating habits and on an increase in physical activity. Greater family involvement was suggested as a potential factor to improve these results, as family involvement was low in all studies.

In order to shed light and evidence on possible strategies to tackle this global health problem, with special interest in young people, the project entitled “Prevention of overweight and obesity among school children in the province of Cádiz: a quasi-experimental study of the effectiveness of a multimodal intervention “PREVIENE-CÁDIZ” [CADIZ-PREVENT] is proposed. The general objective of the study will be to evaluate the effectiveness of the multimodal intervention (individual, family and school) focused on food-related lifestyle habits, physical activity, sedentarism and sleep time, to prevent overweight and obesity in school-aged children (8–9 years) in the province of Cádiz. The main hypothesis of the study is that the intervention will improve eating habits, increase levels of physical activity, reduce screen time and increase sleeping hours of parents and children, as well as reduce the prevalence of overweight and obesity of schoolchildren aged eight to nine years in the province of Cádiz in the long term.

## 2. Materials and Methods

### 2.1. Study Design and Participants

The current study will have a quasi-experimental design with a control group, which will be target children enrolled in third grade of primary education from 26 schools of the province of Cádiz. The inclusion criteria that will be used will be: children belonging to the selected school, who are in the third grade of primary school, regardless of their age, who have the informed consent signed by their parents or legal guardians, who respond to the questionnaires and who undergo to the anthropometric measurements. The exclusion criteria will be that these requirements are not met.

The total number of study participants will be between 850 and 900 children. The parents of each schoolchild will be part of that same subject (father, mother and child are the same subject of study). The sample will be divided into two groups, the intervention group and the control group. The intervention group will contain around 450 schoolchildren and will be made up of 13 of the 26 school centers which will participate in the study. In this group, pre-intervention measurements will be performed, which will be explained later. A multimodal intervention will be developed, which will also be explained in detail later. Finally, post-intervention measurements will be made, which will be the same as the pre-intervention measurements. The control group will also contain around 450 schoolchildren and will be made up of the others 13 school centers which will participate in the study. It will be made up of the same number of schoolchildren. The difference with respect to the intervention group will be that the control group will not receive any educational intervention. Only the pre- and post-intervention measurements will be taken, to later compare the results with the intervention group and find out if an effect has emerged or not.

The province of Cádiz will be divided into four areas: Campo de Gibraltar (six schools), Jerez-Costa Noroeste (five schools), Sierra de Cádiz (nine schools) and Bahía de Cádiz-La Janda (six schools) (Figure 1). In each area, there will be between 200 and 300 participants and school centers belonging to both the intervention group and the control group. This division will be made to have a sample that is as representative as possible of the entire province of Cádiz.

Both the selection of schools participating in the study and the assignment of them to the experimental group or control group will be made randomly from the list of public schools of the Cádiz Education Delegation. These randomizations will be carried out taking into account the schools belonging to each of the geographical areas in which the province of Cádiz will be divided, the socioeconomic condition of the population served, which will be determined from the Material Deprivation Index (MDI) of the section of the census in which the school is located [37], and the number of classes or corresponding lines of each center belonging to that age group. Randomization will be carried out in groups by schools, not individually. 

The group of schoolchildren will constitute the first level of analysis, and the schools will be the second level. This target population (children 8–9 years) has been selected due to the need to detect and solve the problem of childhood obesity in early stages to promote healthier lifestyle habits, and thus achieve health benefits that allow correct weight management from childhood until adulthood, and because, at this age, children are very receptive to receiving information that implies changes in habits [23].

The project, which is 80% co-financed by funds from the Andalusian Operational Program FEDER (European Regional Development Fund) 2014–2020, will conduct in accordance with the Declaration of Helsinki. The protocol has been approved by the Andalusian Biomedical Research Ethics Committee, along with approval from the Delegation of Education and Science of the province of Cádiz (PI-0007-2017, 21 March 2018). In order to guarantee this right, the guarantees of confidentiality of the data will be specified in writing to each family, by means of a letter explaining the objectives of the study, requesting their collaboration and the signed consent of participation, as well as the anonymization of the data. Participation will be voluntary and the collection of information from children will be done by the research team at the school. The questionnaires will be answered in the classroom, and a more intimate space available to the school will be used to carry out the anthropometric measurements, such as a support room, the morning classroom, the library or the gym. Schoolchildren who do not have the informed consent signed by their parents or legal guardians will not participate in the study. These schoolchildren who do not participate in the study and belong to the schools of the intervention group will receive the educational component of the intervention but will not undergo to the pre and post intervention evaluations. And the schoolchildren who do not participate in the study and belong to schools of the control group will not undergo to the pre and post intervention evaluations and of course no intervention. The data obtained in this study will be confidential and will be treated in accordance with Organic Law 3/2018, of December 5, on the Protection of Personal Data, Regulation (EU) 2016/679 of the European Parliament and of the Council, of 27 of April 2016 (General Data Protection Regulation), and in accordance with the official document of the Andalusian School of Public Health on the protection of personal data.

### 2.2. Calculation of the Sample Size

To determine the effectiveness of the intervention on the prevalence of overweight and obesity, the sample size necessary to test hypotheses has been calculated, comparing two proportions. We will compare the percentage of overweight and obesity of schoolchildren before and after the intervention in both groups (intervention and control) (paired data) and inter-group (independent data). In our case, for an estimated difference of proportions of 9% (35% vs. 26%) with a confidence level of 95% and a power of 80%, the minimum sample size required is 410 subjects in each group (intervention and control), which makes a total of 820 subjects. For the calculation of the sample, after reviewing the results obtained by different studies carried out in our environment, it has been assumed an estimated prevalence of overweight and obesity of a 35%, which was found by Villagrán et al. (2013) [9] in a population of Cádiz similar to that of this study.

### 2.3. “PREVIENE-CÁDIZ” [CADIZ-PREVENT] Intervention Protocol

The “PREVIENE-CÁDIZ” [CADIZ-PREVENT] intervention (Appendix A) will have a multimodal nature and will be aimed at schoolchildren, their families and schools. The intervention activities/sessions will be implemented by the teachers at the school centers. The research team will train those teachers to implement the program, and parents in a face-to-face session, but not schoolchildren. To achieve this, before implementing the intervention, the management teams and teachers of the intervention group will receive training sessions managed by the CEP (Center for Teacher Education) of Cádiz from the research team. These sessions will last 10 classroom hours and will be aimed at updating knowledge on promoting healthy lifestyle habits (healthy eating, promoting physical activity, reducing sedentary lifestyle, the importance of sleep and promoting health based on resources). Moreover, the sessions will be addressed for the teaching context, and they will show how to use the tools available to implement the intervention. All the information covered will be made available to the teaching staff through the “Moodle” virtual tool that will allow the research team and teaching staff to be in permanent contact to resolve any possible doubts. There will also be chat tools for management teams and research teams.

The intervention will be implemented in three settings: classroom, family and school. See the description of each setting below.

#### 2.3.1. In the Classroom (Implemented on the Children and Conducted by the Teachers)

It will be a unit of ten health education sessions which will be carried out in the classroom for six months. The teachers in charge of imparting these ten sessions to schoolchildren during the six months will be free to organize the periodicity with which each session is taught, to make it easier for them to combine it with the rest of the content they have to teach. The unit will encompass healthy eating, physical activity habits, reducing sedentary lifestyle and increased sleep, with content based on the healthy eating and physical activity strategies. This content will include (a) nutrition and healthy components of diet and physical activity and the damages of sedentary lifestyle; (b) promotion of the importance of physical activity and participation in sports and recreational activities; (c) review and reduction of the number of weekly hours of sedentary habits; (d) reinforcement of self-concept and body image; (e) importance of sleep and benefits of proper rest; (f) awareness and use of available resources that promote the acquisition of healthy lifestyle habits (Photovoice). Ten sessions will be conducted (Table 1), led by teachers and physical education teachers.

The teachers’ role during the intervention will be planning the timing and distribution of the sessions to be given, as well as conducting these sessions and promoting positive attitudes and opinions regarding food, healthy physical activity, reduction of sedentary lifestyle, and the importance of sleep and use of resources/assets for health available to students. In addition, the Physical Education teachers will promote an activity to improve adherence to physical activity during “Active Recess” breaks.

#### 2.3.2. At the Family Level (with Parents or Caregivers)

The intervention in families will be carried out in three ways: through workshops aimed at parents or caregivers in schools, which will be led by members of the research team based on everyday situations with graphic and audiovisual material. Through a workbook, which will be given by the teachers to the school children, with which parents and children will work together at home to deepen the content of the intervention that the schoolchildren will be seeing in class with the teachers. And through the Photovoice activity, which will consist of taking two photos (jointly at home or outdoors by parents/caregivers and children) one of a resource that encourages healthy eating and another resource that encourages healthy physical activity. Once each child has taken the two photos, they will take them to school and share it, so that all children become aware of the resources they have around them to improve their diet and to promote practice of physical activity. This activity will be explained in detail later.

The workshops aimed at parents given by the research team will take place during one afternoon per school. They will consist of providing parents with skills and strategies to obtain improvements in eating habits, promote healthy meals through cooking workshops, promote the use of health assets available to families, improve the organization and distribution of their children’s free time and improve their children’s sleep habits. This will help children to reduce sedentary habits at home, to increase the practice of sports, recreational and active leisure activities, to improve their eating habits at home and to achieve adequate sleep habits. Table 2 shows how the workshops will be developed and what content each of them will include.

The decision has been made to concentrate all the information in a single afternoon due to the difficulty that parents/caregivers will have to attend school for 3 h in a row, since most of them will work or have obligations and cannot be attending school frequently. In this way, as it is only one day, we will ensure that many more parents attend, that they pay more attention and successfully understand the material. In addition, the information that we intend to give them is not too extensive, so we believe that it can be covered in a single afternoon (3 h). Above all, our intention is to explain what the intervention will consist of, give them basic information on healthy eating, activity physical activity, reduction of sedentary lifestyle and sleep habits, and explain what the exercise booklet and the Photovoice activity consist of, so that they can later spend more time working at home or outdoors with their children.

The role of parents/caregivers during the implementation of the intervention will be to attend the workshops proposed by the research team, to acquire the knowledge that they will later have to share with their children, complete the workbook with them and complete the Photovoice activity. In addition, they will have to encourage healthy eating at home, healthy physical activity, the reduction of sedentary lifestyle and the importance of sleep.

In the case that a family cannot attend the workshop given by the research team, the audiovisual material exhibited in the workshop will be shared with that family. All the information that will be given to the families in the workshop will be summarized in the audiovisual material. Moreover, if any family wants to remember a concept that has been worked on in the workshop, the school will have this material available and will share it to all participating families digitally. In addition, they will be provided with a document through the school in which the nutrition part and the healthy recipes that will be worked on in the healthy cooking workshop will be summarized. This document will be the same that will be provided to families in the workshop. With all this material, families who cannot attend the workshop will be able to acquire sufficient knowledge to successfully exercise their role within the intervention.

#### 2.3.3. At the School Level (with Teachers and the Management Team of the Schools)

In addition to the implementation of the health education unit previously described at the individual level, the intervention in the school environment will involve the main teachers of the participating classrooms, as well as the Physical Education teachers of the centers involved. They will be given training and a guide for teachers will be provided. Moreover, (as indicated in the previous section), the teachers will provide the schoolchildren with a workbook so that they can carry out recreational activities with the family in their time outside of school. Teachers will monitor whether these activities are carried out by reviewing the workbook and, if necessary, by contacting parents. Once the intervention period is over, all participating schools (both those in the intervention group and those in the control group) will be given a report of results and sample teaching guides that they can implement if they wish in subsequent courses.

### 2.4. Procedures and Assessments

#### 2.4.1. Participation of the Schools in Health Promotion Activities

Before the intervention, a questionnaire (Appendix A) created by the research team will be administered to the directors, management teams or teachers of each center. This questionnaire will be used to collect information on whether or not the center currently participates in other health promotion programs, type of program, activities carried out, time spent, among other issues. It is necessary because participation or not in any health promotion program or activity can condition the results of this study. No school will be excluded from the study if it participates in a health promotion program, regardless of whether it belongs to the intervention group or the control group. What will be done, when the analysis is carried out, will be a differentiation into several categories, taking into account in a differentiated way the centers that participate in some of these programs from centers that do not.

#### 2.4.2. Sociodemographic and Socioeconomic Information

Two questionnaires (Appendix A) will be used to record the sociodemographic and socioeconomic information of both schoolchildren and family members. The questionnaire administered to schoolchildren (Appendix A) will include questions about the number of people they live with at home, who they live with, among others. The questionnaire addressed to parents (Appendix A) will include questions related to the place of birth, educational level, employment situation and occupation, income level, among others. In this way, the sociocultural situation of each participating family will be assessed.

This information will be recorded both before and after the intervention. Before, in the pre-intervention measurements, which will be carried out during the two months before starting the intervention. And later, in the post-intervention measurements, which will take place during the two months after the intervention.

These questionnaires have been adapted from the study “Previene” [Prevent] carried out in Granada [10] and the study “Prevention of childhood obesity in Barcelona” (POIBA) carried out in Barcelona [38], since our project is based on the experiences of these previous studies. The results of the study carried out in Granada have not yet been published, so it is not yet known whether the intervention was effective or not. In the case of Barcelona, the Public Health Agency has published results of the intervention [38], showing that the program is capable of preventing one out of every 3 new cases of childhood obesity, improving eating frequency and behavior, practicing physical activity, habits sedentary and knowledge.

As a result of these interventions, certain aspects will be corrected and the present one will be improved. The main differences will be that in “PREVIENE-CÁDIZ” [CADIZ-PREVENT] more variables will be included in the pre- and post-intervention measures (evaluation of physical activity, physical condition, sedentary lifestyle, breastfeeding, type of childbirth and birth weight). The intervention will be improved, including activities such as “Photovoice”, which will be explained later. Two tools will be included to improve the evaluation of the satisfaction/reaction of schoolchildren, family members, teachers and management teams. These tools will be the “World Cafe”, which will be explained in detail later, and a questionnaire for teachers and management teams. All these aspects were not considered in the previous interventions and have been considered relevant by the research team to include them in this one.

#### 2.4.3. Anthropometric Measurements

Measurements will be taken both before and after the intervention by previously trained research team members, following the standardized International Society for the Advancement of Kinanthropometry (ISAK) procedure [39]. ISAK is an international society dedicated to the study of kinanthropometry, which has created an internationally standardized procedure for taking anthropometric measurements of children and adults.

Body weight will be measured with a mechanical scale sensitive to 100 g (SECA Colorata 760, Hamburg, Germany). Height will be measured using a portable stadiometer with a precision of 0.1 cm (SECA 213, Hamburg, Germany). Body Mass Index (BMI) will be calculated as weight in kilograms divided by height in meters squared and will be used to determine the level of overweight or obesity. If the BMI of the students is between 18.44 and 19.84 according to Cole’s cut-off points [40], they will be considered to be overweight. If the BMI is between 21.60 and 24.00 [40], they will be considered obese. In addition to this formula, the BMI will also be calculated by estimating the percentiles according to their age based on the World Health Organization recommendations.

The skinfold thickness of the triceps and subscapularis will be measured with a caliper with a precision of 0.2 mm (HOLTAIN DIM-98.610ND, London, UK) and will be used to estimate both the percentage of fatty tissue and muscle tissue. For the calculation of the percentage of fatty tissue, the regression equation proposed by Boileau, Lohman and Slaughter [41] will be used. The skinfold thickness of the triceps will also be used as an indicator (proxy) of obesity, considering values greater than 20 mm as obesity, with a sensitivity of 0.79 for the age range of our study [42]. Furthermore, the waist and hip circumferences will be measured with a 1 mm precision tape measure (SECA 201, Hamburg, Germany) and the waist-hip ratio will be calculated. In addition, the waist/height ratio will be calculated as an indicator related to childhood obesity [43].

#### 2.4.4. Self-Reported Physical Fitness

The self-reported physical fitness of children (Appendix A) and parents (Appendix A) will be registered using The International Fitness Scale (IFIS), previously validated [44,45], before and after the intervention. This scale evaluates both overall physical condition as well as each one of its primary components specifically: cardiorespiratory fitness, muscular strength, speed-agility and flexibility.

#### 2.4.5. Physical Activity and Screen Time

Physical activity and screen time will be recorded by means of a self-report by the schoolchildren (Appendix A) and observation of the parents using a built-in questionnaire (Appendix A) from the one used in the “Previene” [Prevent] study in Granada [10]. Information will be collected on the amount of physical activity and screen time (TV, computer, video games…) of schoolchildren both before and after the intervention. In addition, the knowledge, attitudes and beliefs about physical activity, sports and sedentary lifestyle presented by both students and families will be evaluated.

#### 2.4.6. Diet

The daily/weekly consumption of processed baked goods, fried foods, snacks, sugary soft drinks, fruits, vegetables, packaged juices, sandwiches, dairy products, carbohydrates, meat, cold cuts, fish, and vegetables, among others, will be recorded through questionnaires [10] applied to schoolchildren (Appendix A) and family members (Appendix A) both before and after the intervention. In addition, the knowledge/beliefs about food and diet of schoolchildren and relatives, and the use of the school canteen and its evaluation [10] will be assessed. Schoolchildren will answer questions about all of these variables about themselves, and parents will answer both about themselves and about their children.

#### 2.4.7. Sleep, Overall Health, Feelings and Emotions

Schoolchildren’s daily hours of sleep, both for Monday to Friday as well as for weekends, will be recorded through questionnaires (Appendix A) adapted from those used in the “Previene” [Prevent] study in Granada [10]. Parents will be asked about their children before and after the intervention. Parents will have to register the time their children normally go to bed and get up. Furthermore, the general health status, feelings and emotions of the students will be recorded using a specific questionnaire (Appendix A) also adapted from the “Previene” [Prevent] study [10] applied before and after the intervention.

#### 2.4.8. Health Resources

Health resources will be evaluated using the Photovoice qualitative technique [46]. This technique will consist of taking two photographs, one of a resource that encourages healthy eating and the other from a resource that encourages healthy physical activity. Once each family identifies two health resources, the students will return to work on it in the classroom, which means that this activity will involves two sessions in the classroom and at least one with the family. This technique will be used for a dual purpose: (1) to identify the main health resources/assets related to physical activity and diet available to families in the environment of each school; (2) for educational purposes, as this technique will be part of the activities of the intervention, with which it is intended that both schoolchildren and family members identify, become aware of and promote the use of previously identified resources/assets.

#### 2.4.9. Breastfeeding, Childbirth Type and Birth Weight

The characteristics of breast milk intake, type of delivery and birth weight will be recorded after the intervention using a questionnaire for parents (Appendix A), which has been created by the research team based on the information from the scientific literature determined as important about these variables related to childhood obesity [47,48,49,50,51].

#### 2.4.10. Participant Satisfaction/Reaction and Adherence to the Intervention

Once the intervention has been carried out, the participants will be asked about their perception of the program, the main health resources that have been discovered or identified, the strengths and weaknesses of the intervention and any aspects to be improved. The research team will record the answers to these questions to assess the adherence of the participants to the intervention, including children´s adherence, caregivers´ adherence and teachers’ adherence.

These aspects will be studied using two different activities. First, a qualitative technique called “World Cafe” [52] which will consist of gathering a sample of the participating agents (teachers, management teams, parents and schoolchildren) and dividing them into three different groups: (1) teachers and management team, (2) parents, and (3) schoolchildren. Each of the groups will be accompanied by someone from the research team, who will act as a moderator of the table, guiding the conversation through a script of semi-structured questions.

There will be 3 rounds and two changes. Each round will consist of having a conversation between the members of each group about the questions of the semi-structured script. Once all questions in the first-round semi-structured question script are completed, there will be a change. This change will consist in that a member of group 1 will go to group 2, a member of group 2 to group 3 and a member of group 3 to group 1. In this way all groups will have a new member of another group, and it will start a new round following the script of semi-structured questions. Once this round is completed there will be another change, but this time a member of group 1 will go to group 3, a member of group 2 to group 1, and a member of group 3 to group 2, to get each group to have at least one member from each one of the 3 groups, to achieve a multi-group exchange of ideas. Then the last round will begin again following the semi-structured question script, and once it is finished, the main ideas that have emerged from the conversation will be shared. A member of the research team, who will act as moderator of the session, will be in charge of making the decision of when to change rounds and of making the changes.

The aim of this technique will be to know the satisfaction of the participants with respect to the intervention, and also to assess the adherence to the intervention. It will be intended to know if the intervention has achieved real change in the home of the schoolchildren regarding better eating habits, increased levels of physical activity, reduction of sedentary lifestyle and sleep improvement (children´s adherence and caregivers’ adherence), and also to know how the program has gone in the schools (teachers´ adherence).

Qualitative methodology will be used for the analysis of the World Cafe interviews. A content analysis will be carried out on all the information obtained. Both the verbal information (of which an exhaustive transcription will be made), the reports of the table coordinators, the photographic images, as well as the written and iconographic material generated. The development of the sessions, the moment, context and origin of the ideas that arise will always be taken into account. When considering the World Cafe as a semi-structured group interview technique, opinions on development and general effectiveness, the health assets that have been identified, the main strengths and the elements that should be improved in the program will be studied, but within a flexible framework in which all emerging elements will be considered.

Secondly, in addition to the World Cafe, in order to know the satisfaction/reaction and adherence to the intervention of teachers and management teams, a questionnaire will be administered to teachers (Appendix A) and management teams (Appendix A) participating in the study. They will be asked about the development/implementation of the program (whether they carried it out or not, time spent in developing it, dynamics, etc.), as well as if they have modified or supplemented it, and any possible proposals for improvement. These questionnaires will be designed by the research team of this study.

Table 3 lists all the measurements that will be collected, the tools with which they will be measured, and when they will be measured.

### 2.5. Statistical Analysis

A descriptive analysis of the main outcome variables will be carried out with measures of centrality, dispersion and 95% confidence intervals (CI), and percentages or prevalence for the categorical variables. Chi square tests will be used for categorical variables and *t*-test or ANOVA for continuous variables, which will be used for comparisons of percentages or means, respectively, when they have a normal distribution. The effectiveness will be studied by comparing the changes in the groups. Simple and multiple linear regressions and/or logistic regressions with the respective odds ratios (OR) and the 95% CI will be applied as an association measure. The grouping term will be modeled in the analysis to adjust the effect of the intraclass correlation and a multilevel test will be conducted. For this, several levels of grouping will be established. The first level will be constituted by the geographical areas of study, the second by the municipality, the third level of grouping will be the school center and the line or classroom, and the fourth and last will be given by the intervention (control and experimental group).

For the analysis of longitudinal data, MANOVA of maximum likelihood and least squares will be used under conditions of normality of the data (assuming that large sample sizes will be obtained). If the data are not normal, generalized linear models (GLM) will be used. As a strategy for the analysis of missing data, the analysis of missing values of the SPSS statistical package will be used. In most situations it is not possible to verify the association of the missing data with other variables that allow it to be used to impute them in a probabilistic way; the Little’s MCAR test will be used to determine if the missing data occurs completely randomly (“Missing Completely at Random Data: MCAR”) and if its imputation is necessary. If the value of p obtained is less than 0.5, it cannot be assumed that the data are completely lost by chance and a multiple imputation will be made.

The dependent variables of our study will be BMI, skinfold thickness of the triceps, waist/hip ratio (as anthropometric variables) and the prevalence of overweight and obesity. As surrogate or intermediate variables we will analyze behaviors related to eating, physical activity and sedentary lifestyle, hours of sleep, knowledge and beliefs about eating and physical activity, as well as the physical condition (IFIS) [44,45] of the children.

In the models, possible covariates related to personal, family, school and environment characteristics will be analyzed. Especially will be analyzed: sex, birth weight, breastfeeding, sociodemographic characteristics, family socioeconomic level, level of physical activity and time of sedentary activities of the parents; behavior, knowledge and beliefs about family food and physical activity, participation of the school in the intervention and level of adherence, participation of the center in other health promotion programs, availability or use of the school canteen and the existence of health assets in the environment.

To determine the efficacy of the different components of the intervention, predictive models will be obtained for each of the outcome variables by backward multiple linear regression, in which all possible components are included and are progressively eliminated from lowest to highest contribution. To analyze the specific contribution of a certain component on a dependent variable, the semi-partial correlations will be obtained; this method allows the effect of the component to be controlled to be subtracted from the predictors being treated. The R^2^ values of the components measure their theoretical contribution and allow establishing a theoretical hierarchy among the components that will be used in the regression by blocks; by changing R^2^ the different contributions of the variables will be seen.

### 2.6. Timeline of the Intervention

With the intention of covering the proposed objectives, the intervention will be conceived as a gradual process in which three phases will be able to be distinguished fundamentally: (I) preparation phase, (II) implementation phase and (III) evaluation and dissemination phase.

Phase I will essentially cover the first year of the project. It will include activities such as the search and updated review of the scientific literature, preparation of the necessary material to carry out the evaluations and the intervention (questionnaires, educational program, booklets), consensus meetings between the research team, recruitment and meetings with the management teams and/or teachers from the centers participating in the study, recruitment of participating teachers and agreements on the monitoring system during the intervention, design and implementation of teacher training and design of workshops for families.

Phase II, which will be the implementation phase, will cover the second year of the study. It will include activities such as conducting pre and post intervention evaluations, implementation of the multimodal intervention, realization of workshops for families, creating databases and entering data. In addition, it will include activities such as the identification of resources/assets for health related to healthy eating and physical activity in the environment of each school, through the Photovoice technique. In turn, it will include the implementation of the World Cafe technique, which will be used to describe the perception of the participants of the intervention (schoolchildren, teachers and parents). Finally, it will include the identification of the current prevalence of childhood overweight and obesity in the province of Cádiz.

All schools will have the same time to implement the intervention (6 months). Within these 6 months, each school will have to combine its usual activities with the implementation of the educational program. To guarantee that all the schools carry out all the sessions, the research team will schedule the school calendar with the teachers and management teams of the participating centers in the teacher training workshops that will be carried out in phase I. The research team will provide to the schools all the necessary resources to implement the educational program.

Phase III, which corresponds to the evaluation and dissemination phase, will basically cover the third and final year of the study. It will include activities such as the statistical analysis of the data and disseminating results. The results will be disseminated both to the scientific community (through scientific papers, congresses, conferences, etc.) and to the participating schools (through a report in which the main results of the study will be specified). In addition, it is also intended to disseminate the results to society in general, so that the relevant authorities become aware of the problem and, if it is possible the intervention is implemented (with all the relevant modifications) in as many schools as possible.

## 3. Discussion

“PREVIENE-CÁDIZ” [CADIZ-PREVENT] intervention is based on the scientific literature on strategies to reduce childhood overweight and obesity. The design of the most effective interventions for the prevention of childhood obesity must include an education program carried out in the classroom and at home, where children and parents work aspects such as the practice of regular physical activity [53], recreational activities [54], extracurricular sports [35], reduced screen time, reduced fat consumption, increased fruit and vegetable servings and decreased consumption of carbonated beverages [54], in order to improve the behavior of children and parents and get them to acquire healthier lifestyle habits [29].

According to a Cochrane Review [29], most interventions take place within the 6–12 year range. Solid evidence has been found regarding the benefits which can be obtained within those ages [29]. This result is consistent with a later review [29] which concludes that those interventions combining physical activity and diet were those which yielded the best results both in their family settings and in school. Taking this into account, it can be very interesting to delve deeper into this age range, focusing on third grade of Spanish Primary Education due to the maturity of the students to be able to fill in questionnaires and to understand the intervention. Due to the fact that this intervention will take place through sessions included in their ordinary curriculum, it is necessary to pick a subject to develop such material and to implement it on the same level in all schools, considering that such school year is the best one to do that. In addition, this age has been chosen by other authors for different interventions on health promotion [25,55].

Among the strengths of this project, we highlight that a multimodal intervention will be carried out incorporating the main agents responsible for education (school and family) on a representative sample of boys and girls aged 8–9 years from the province of Cádiz. In addition, the multidisciplinary composition of the research team with more than 10 different disciplines and the continuous feedback and workshops (seminars) between the research team and the teachers will be strengths of the PREVIENE-CÁDIZ [CADIZ-PREVENT] project.

Regarding sustainability, it is worth mentioning that the initiative could be extended without the need for considerable additional resources. The fact that the educational sessions are implemented by the teachers enables an easier continuity of the intervention. 

As a limitation, getting all the schools in the intervention group to carry out the ten sessions on the planned schedule will be a challenge for the project. For this limitation, a contingency plan is contemplated whereby, through the teacher training courses developed by the research team in the CEP of Cádiz, a section will be included to organize the scholar calendar with the management teams and teachers. This will ensure that all schools commit to implementing the intervention at the same time, and it is expected that this will favor the implementation of the sessions.

In another vein, not measuring the physical condition of schoolchildren and family members objectively, but doing so subjectively using the self-report questionnaire of perceived physical condition (IFIS), could be a limitation. However, due to the complexity of bringing together schoolchildren, and especially their families to perform a series of physical tests, a decision has been made to use the IFIS scale, which has previously been validated and used in this population group [44,45].

## 4. Conclusions

This study will enable the evaluation of the effectiveness of an educational multimodal intervention at the individual, school and family levels, focused on the acquisition of healthier lifestyle habits with regard to diet, increased levels of physical activity, reduced sedentary lifestyle and improved sleep to prevent overweight and obesity in a representative sample of schoolchildren from 8 to 9 years old in the province of Cádiz. Once the evaluation has been carried out, the weaknesses and strengths of the educational program with respect to the population of the third year of primary education as a target group will be able to be identified, and therefore, timely decisions will be able to be made to adapt, modify and improve the educational intervention.

The results of this project will allow for the creation of well-adapted educational programs for this child population, which will be able to serve as a reference for the implementation of similar programs in other locations. Once the factors that will influence the possible effectiveness of the educational intervention have been determined, its implementation strategy (focused, selective or adapted implementation) will be able to be improved according to the personal, behavioral, family, and contextual characteristics of the relevant target population.

## Figures and Tables

**Figure 1 ijerph-18-01622-f001:**
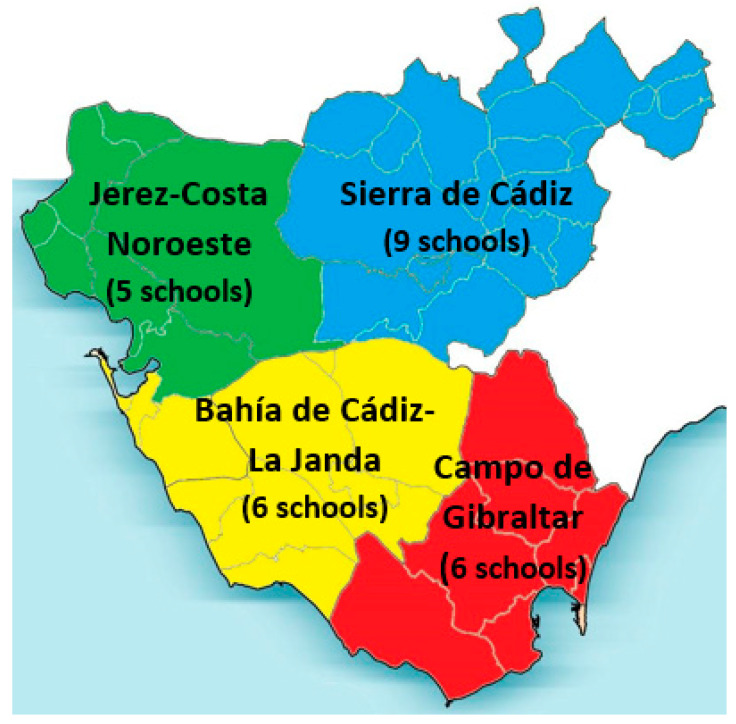
Geographical representation of the 4 study areas in the province of Cádiz.

**Table 1 ijerph-18-01622-t001:** “PREVIENE-CÁDIZ” [CADIZ-PREVENT] intervention sessions aimed at schoolchildren.

Sessions	Objectives
Session 1 Our development	To recognize weight and height as signs of development.
Session 2 Let’s assess	To recognize and accept one’s own physical appearance and that of others.
Session 3 Our digestion	To get to know the digestive tract and its functions; to practice dietary hygiene habits.
Session 4 Food groups	To get to know the food groups and how to differentiate them according to their origin.
Session 5 Nutrients and their functions	To get to know nutrients and their functions.
Session 6 The food pyramid and exercise	To get to know the primary recommendations for eating and exercising; to show skills associated with these recommendations.
Session 7 The best breakfast	To identify content and quantities of a health breakfast; to experiment how breakfast can be attractive and desirable.
Session 8 Physical activity and sleep	To Improve health habits of physical activity, sleep and eating.
Session 9 Review and synthesis	To integrate physical activity, sleep and healthy eating into habits.
Session 10 Photovoice	To identify, to become aware and to use the health resources readily available.

Table adapted from the “Previene” [Prevent] study carried out in Granada [10].

**Table 2 ijerph-18-01622-t002:** Development of workshops aimed at parents or caregivers.

Workshops	Length	Content	Materials
Healthy eating workshop	45 min	Macronutrients (carbohydrates, fats and proteins) Micronutrients (minerals and vitamins) Nutrition labeling Developing a healthy monthly menu	Graphic and audiovisual media
Healthy cooking workshop	45 min	Making healthy bread: the recipe for healthy bread will be given to parents and the bread will be made in front of them so they can see how it is made. They will take a previously prepared bread so they can taste it. Making healthy cookies: parents will be given the recipe for healthy cookies and cookies will be made in front of them so they can see how they are made. Pre-made cookies will be taken so they can try them.	Two glass bowls A whisk
Healthy lifestyles workshop	45 min	What is physical activity? Benefits of healthy physical activity Tricks to increase the time of healthy physical activity Harms of sedentary lifestyle Tricks to reduce sedentary lifestyle Planning of free or leisure time	Graphic and audiovisual media
Workshop on assets/resources available to improve health	45 min	Current health concept What are assets for health? What resources/assets for health exist? Resources/assets for health that we have at our disposal Motivation to improve our health status	Graphic and audiovisual media

**Table 3 ijerph-18-01622-t003:** Measures that will be collected in the study.

Measure/s	Tool/s	Measurement Time
Participation of the schools in health promotion activities	Questionnaire addressed to directors, management team or teachers Name: Questionnaire—Cádiz Prevent Program: Directors—Management teams—Teachers (Appendix A) Number of items: 6 Reference: modified from the intervention carried out in the study “Previene” [Prevent] from Granada [10] and POIBA from Barcelona [38]	Within 6 months before the intervention
Sociodemographic and socioeconomic information	Questionnaire addressed to schoolchildren Name: How we eat (Appendix A) Number of items: 8 Reference: modified from the intervention carried out in the study “Previene” [Prevent] from Granada [10] and POIBA from Barcelona [38] Questionnaire addressed to family or legal guardians Name: Families. How we eat and move (Appendix A) Number of items: 11 Reference: modified from the intervention carried out in the study “Previene” [Prevent] from Granada [10] and POIBA from Barcelona [38]	Pre intervention measurements: within 2 months before the intervention and post intervention measurements: within 2 months after the intervention
Anthropometric measurements	Body weight: mechanical scale sensitive to 100 g (SECA Colorata 760, Hamburg, Germany) Height: portable stadiometer with a precision of 0.1 cm (SECA 213, Hamburg, Germany) Triceps skinfold thickness and subscapular skinfold thickness: caliper with a precision of 0.2mm (HOLTAIN DIM-98.610ND, London, England) Waist and hip circumference: tape measure with a precision of 1mm (SECA 201, Hamburg, Germany)	Pre intervention measurements: within 2 months before the intervention and post intervention measurements: within 2 months after the intervention
Self-reported physical fitness	Questionnaire addressed to schoolchildren Name: International Fitness Scale (IFIS) (Children version) (Appendix A) Number of items: 5 Reference: PROFITH research group, Granada, Spain [44] Questionnaire addressed to family or legal guardians Name: International Fitness Scale (IFIS) (Adults version) (Appendix A) Number of items: 5 Reference: PROFITH research group, Granada, Spain [45]	Pre intervention measurements: within 2 months before the intervention and post intervention measurements: within 2 months after the intervention
Physical activity and screen time	Questionnaire addressed to schoolchildren Name: How we move and feel (Appendix A) Number of items: 17 Reference: modified from the intervention carried out in the study “Previene” [Prevent] from Granada [10] and POIBA from Barcelona [38] Questionnaire addressed to family or legal guardians Name: Families. How we eat and move (Appendix A) Number of items: 15 Reference: modified from the intervention carried out in the study “Previene” [Prevent] from Granada [10] and POIBA from Barcelona [38]	Pre intervention measurements: within 2 months before the intervention and post intervention measurements: within 2 months after the intervention
Diet	Questionnaire addressed to schoolchildren Name: How we eat (Appendix A) Number of items: 56 Reference: modified from the intervention carried out in the study “Previene” [Prevent] from Granada [10] and POIBA from Barcelona [38] Questionnaire addressed to family or legal guardians Name: Families. How we eat and move (Appendix A) Number of items: 91 Reference: modified from the intervention carried out in the study “Previene” [Prevent] from Granada [10] and POIBA from Barcelona [38]	Pre intervention measurements: within 2 months before the intervention and post intervention measurements: within 2 months after the intervention
Sleep, overall health, feelings and emotions	Questionnaire addressed to schoolchildren Name: How we move and feel (Appendix A) Number of items: 22 Reference: modified from the intervention carried out in the study “Previene” [Prevent] from Granada [10] and POIBA from Barcelona [38]	Pre intervention measurements: within 2 months before the intervention and post intervention measurements: within 2 months after the intervention
Health resources	Photovoice Addressed to schoolchildren and family or legal guardians Reference: Wang et al. [46]	During the intervention (6 months)
Breastfeeding, delivery type and birth weight	Questionnaire addressed to family or legal guardians Name: CADIZ-PREVENT—Breastfeeding, childbirth and birth weight (Appendix A) Number of items: 7 Reference: created by the research team based on the information from the scientific literature [47,48,49,50,51]	Post intervention measurements: within 2 months after the intervention
Participant satisfaction/reaction and adherence to the intervention	World Cafe Addressed to schoolchildren, family or legal guardians, teachers and management teams of four schools selected of the intervention group Reference: www.theworldcafe.com Questionnaire addressed to management teams of the intervention group Name: CADIZ-PREVENT Program Evaluation Questionnaire. Directors—Management teams (Appendix A) Number of items: 14 Reference: designed by the research team of the CADIZ-PREVENT study Questionnaire addressed to teachers of the intervention group Name: CADIZ-PREVENT Program Evaluation Questionnaire. Teachers (Appendix A) Number of items: 35 Reference: designed by the research team of the CADIZ-PREVENT study	Post intervention measurements: within 2 months after the intervention

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
