# Peer review of "A Multimodal Intervention for Prevention of Overweight and Obesity in Schoolchildren. A Protocol Study “PREVIENE-CÁDIZ”"

_ijerph, 2021, doi:10.3390/ijerph18041622_

Round 1
Reviewer 1 Report
- Page 3, line 122: Was “low” involvement of the family meant? It seems counter-intuitive that low family involvement would be associated with the positive changes in eating habits and activity.
- Given the small number of schools to be included in the study, will methods for randomizing schools to interventions ensure balance of the interventions across geographical areas or other relevant characteristics such as socioeconomic characteristics of the student population? If not, what are the potential ramifications of a lack of balance? Would that be a weakness of the study?
- Is there prior evidence to support that a 10-week intervention and final assessments 2 months later allows enough time to see a change in overweight or obesity percentages from 35% to 26% (assumptions used in the sample size calculation)?
- It is not clear whether the sample size estimates account for the cluster randomized design of the study (schools randomized to interventions). Please explain how it does or state that it does not.
- What specific guidance will be presented in the intervention regarding the definition of a healthy diet, target levels of activity, hours of sleep, etc., and what evidence supports these recommendations?
- In the statistical analysis section, please describe how the grouping term will be included in the model in order to model the intraclass correlation (for example, will a random effect be used). Does “grouping” refer to school? Please also explain further what is meant by “a multilevel test will be conducted.”
- If the intervention is being delivered at the classroom level then that would be an additional level of clustering in the study design. Will that be accounted for in analysis?
- It is not clear to me how the efficacy of the different components of the intervention can be estimated. If all components of the intervention are given together as a package, then how can the effect of each be estimated separately?
- How is adherence to the intervention defined?
- Please state the aims of the analyses that will include covariates at the child-level, or school-level.
- Will the characteristics of children and families who consent to the study be compared to those who do not consent to see if there is bias in enrollment?
Author Response
Dear reviewer,
We leave you attached in word point-by-point the details of the revisions in the manuscript and the responses to your comments.
Thank you very much for giving us the opportunity to rewrite and resubmit this manuscript.
Sincerely,

Reviewer 2 Report
Thank you for the opportunity to review the manuscript entitled, "A Multimodal Intervention for the Prevention of Overweight and Obesity Among Schoolchildren in the Province of Cádiz “PREVIENE-CÁDIZ”. Protocol Study”.
I believe this study investigates a topic relevant to the readers of “IJERPH”. My opinion is that in the current form the manuscript does not have scientific quality to be published in IJERPH. The manuscript is far below scientific standards and lacks clear structure, participants, instruments, results, discussion…
It is an investigation that is not developed. This is not a scientific article.
I encourage the authors to submit it for publication when they finish the study.
Author Response
Response to Reviewer 2 Comments
Point 1: I believe this study investigates a topic relevant to the readers of “IJERPH”. My opinion is that in the current form the manuscript does not have scientific quality to be published in IJERPH. The manuscript is far below scientific standards and lacks clear structure, participants, instruments, results, discussion…
It is an investigation that is not developed. This is not a scientific article.
I encourage the authors to submit it for publication when they finish the study.
Response 1: We regret that the manuscript does not meet your expectations for publication in the journal. In any case, we will continue to publish results of the study when it is carried out. Thank you very much for your comments and recommendations.
Reviewer 3 Report
Statistical analysis section lists a series of tests, but no specific hypotheses/research question that will guide the particular analyses. Please include the research questions/hypotheses that will guide the study.
Here are some additional suggestions for edits:
Abstract
Line 29---Suggestion: This study examines the need to address the high prevalence (40%) of…..
Line 33---and an increase in health spending. In this context, it is necessary to intervene promoting….
Line 34---The objective of the project was to evaluate the effectiveness……
Line 36----The intervention focused mainly on diet,…..
Line 38----It consisted of a 10 session education program
Line 39----In addition, children had two workbooks, …
Line 40---In addition, children had two workbooks, …..
Line 41-----To do this, a study with a quasi-experimental design with a control group was carried out.
Line 70-----This data confirm the need to develop and reinforce public health policies that reduce these numbers.
Line 96---proposed, since this type of intervention allows individuals to overcome these inequalities…..
Line 112---intervention lasted six months.
Line 121……that half of the studies showed positive changes in the improvement of body composition, and all had…….
Line 141….legal guardians, and who, in addition to the questionnaires, have undergone the anthropometric…..
Line 149- The control group will consist of the others 13 school centers…
Line 156--In each area, there will be between 200 and 300 participants.
Line 170--The group of schoolchildren will constitute the first level of analysis, and the schools will be the second level.
Lines 171-172--- This target population (children 8-9 years) has been selected due to the need to detect and solve the problem of childhood obesity in early stages to promote healthier lifestyle habits, and thus……
Line 174-- ….because, at this age, children are very receptive…..
Line 185---… answered in the classroom, and a more intimate space available to the school will be used to carry out…..
Line 206…. which was found by Villagrán et al., (2013) [9] in a…..
Line 298….. Before the intervention, a questionnaire created by the research team will be administered to the…..
Line 309-- The questionnaire administered to schoolchildren will…..
Line 312-- … In this way, the….
Line 544-- Among the strengths of this project, we highlight the realization of a multimodal intervention…
Line 559-- In another vein, not measuring the physical condition of schoolchildren and family members objectively,…..
Author Response

(The authors gave the same response as above.)

Reviewer 4 Report
The Article “A multimodal intervention for the prevention of overweight and obesity among schoolchildren in the province of Cádiz “PREVIENE-CÁDIZ”. Protocol study” is well and quite clearly written but it requires a number of changes before it will be published. The major strength of the study is this study practical application. There are some remarks concerning this article:
- The title of article has to be corrected – it is too long and not clearly representing the aim of the study.
- In the article it was mentioned the length of intervention – 10 sessions, but it is not clear is it each week, each month or etc. All the sessions would be together for the parents and schoolchildren? Moreover, it would be valuable to prove scientifically in the introduction the length of the intervention.
- Some terms usage could be corrected, e.g. “after-school program” and etc.
- The number of involved in the research participants not very clear and in some places presented differently (e.g. lines 143-145 and 156). And it is not related to sample size calculation (line 203).
- In 1 table presented intervention program not clarified whom it addresses.
- It is not clear how will be controlled recreational activities of schoolchildren in time outside the school (lines 291-292).
- Some intervention activities not clear described, e.g. “Photovoice”.
- BMI for children is better to estimate the percentiles according to their age.
- Why physical fitness will be estimated using self-reported scale? I don’t think that such an age schoolchildren will be able to it themselves.
Before publication, in my opinion, article must be improved.
Author Response

(The authors gave the same response as above.)

Round 2
Reviewer 1 Report
Thank you for the responses and edits which have provided clarification on the methods used in this study. My concerns have been addressed.
Author Response
Dear reviewer,
Thank you very much for reviewing our manuscript again and considering it for publication.
I attach a Word document with the responses to your comments.
Thank you very much for everything,
Sincerely,
Ruben.

Reviewer 2 Report
Thank you for the opportunity to review again the manuscript entitled, "A Multimodal Intervention for the Prevention of Overweight and Obesity Among Schoolchildren in the Province of Cádiz “PREVIENE-CÁDIZ”. Protocol Study”.
The manuscript does not meet my expectations for publication in the journal because is not finish.
I encourage the authors again to submit it for publication when the results are obtained.
Thank you very much.
Author Response
Dear reviewer,
Thank you very much again. We regret that our manuscript does not meet your expectations for publication in the journal. Following your recommendations, we will send new articles to the journal when the study is finished.
Thank you very much for everything anyway.
Sincerely,
Rubén.